# Aberrant Left Subclavian Artery-Esophageal Fistula in a Patient with a Prolonged Use of Nasogastric Tube: A Case Report and Literature Review

**DOI:** 10.3390/diagnostics11020195

**Published:** 2021-01-28

**Authors:** Sungbin Kim, Kyung Nyeo Jeon, Kyungsoo Bae

**Affiliations:** 1Department of Radiology, Institute of Health Sciences, Gyeongsang National University School of Medicine, Jinju 52727, Korea; kmsgbn0510@gmail.com (S.K.); ksbae@gnu.ac.kr (K.B.); 2Department of Radiology, Gyeongsang National University Changwon Hospital, Changwon 51472, Korea

**Keywords:** esophageal fistula, subclavian artery, bleeding, nasogastric, intubation

## Abstract

Arterial-esophageal fistula is a rare but potentially fatal complication. Right aortic arch with aberrant left subclavian artery is a rare congenital vascular anomaly that can cause esophageal compression, particularly when the proximal portion of the aberrant subclavian artery forms a Kommerell’s diverticulum. Prolonged use of a nasogastric tube can cause pressure necrosis of the esophagus. We report a patient with massive gastrointestinal bleeding secondary to aberrant left subclavian artery-esophageal fistula after a prolonged use of nasogastric tube. A high index of suspicion is essential for better prognosis when a patient with congenital aortic arch anomaly shows upper gastrointestinal hemorrhage.

## 1. Introduction

Although arterial-esophageal fistulae are rare, they can cause massive and life-threatening hemorrhage. Timely diagnosis and treatment are mandatory, since these fistulae are fatal in a majority of cases [1,2]. Even in adults, it is important to recognize congenital variants and anomalies of the aortic arch, as these anomalies have important implications for surgical and percutaneous interventions [3]. The embryological origin of aortic arch anomalies is based on the hypothetic double aortic arch [4]. Depending on differences in the regression segment, various types of aortic arch anomalies can develop. If the regression occurs between the left common carotid artery and the left subclavian artery in the left arch, the right-sided arch with an aberrant left subclavian artery (ALSA) will be formed (Figure 1).

In this anomaly, ALSA takes a retroesophageal course. Its proximal portion often forms a dilated segment called the diverticulum of Kommerell, thus compressing the esophagus posteriorly. In addition to a transmural pressure from the adjacent artery, a prolonged use of nasogastric (NG) tube may contribute to arterial-esophageal fistula formation due to the creation of pressure necrosis where the tube contacts with the esophageal wall [5].

We report a patient with massive upper gastrointestinal bleeding secondary to ALSA-esophageal fistula after a prolonged use of NG tube. We reviewed the associated literature and addressed a few critical issues that physicians should be aware of.

## 2. Case Report

This study was approved by the Institutional Review Board of our hospital. The requirement of informed consent was waived. A 63-year-old man was transferred from a nursing hospital for low blood pressure (76/51 mmHg) and bloody gastric tube drainage. His red blood cell count was 2.44 × 106/μL, and his hemoglobin level was 7.4 g/dL. The patient was in a bed-ridden state for a year due to sequelae of traumatic intracranial hemorrhage. He was on hemodialysis for chronic renal failure. He also had old myocardial infarction and diabetes. A radiograph with an anteroposterior view of the chest showed the right-sided aortic arch (Figure 2). Upper GI endoscopy revealed about a 2-cm-wide mass-like lesion with an ulcer and red pigmentation in the upper esophagus, at a distance of 23 cm from the incisors (Figure 3a). Atrophic mucosal change was seen at the gastric body and the antrum. Active bleeding focus was not found. To exclude the possibility of an esophageal cancer, a biopsy was performed for the esophageal lesion. After supportive treatment, the patient was sent back to the nursing hospital on the same day.

Two days later, the patient revisited our hospital because he could not get hemodialysis due to hypotension (systolic pressure, 60 mmHg) and decreased thrill on the arteriovenous fistula. Laboratory data demonstrated a drop in hemoglobin values (5.1 g/dL) and leukocytosis (white blood cell count: 12 × 10^3^/μL). On upper GI endoscopy, an actively bleeding and pulsating vessel was seen in the previous esophageal ulcer base (Figure 3b). The pathology specimen obtained from the esophagus two days before did not show any evidence of malignancy.

Computed tomography (CT) revealed a right-side aortic arch with ALSA. ALSA took a retroesophageal course. Its proximal portion formed a diverticulum of Kommerell. The esophagus was compressed by the diverticulum of Kommerell (Figure 4). On sagittal reconstruction image, bleeding from the diverticular part of ALSA to esophagus was suspected (Figure 4c). Other parts of the chest and abdomen were unremarkable.

Thoracic arteriography confirmed the extravasation from an aneurysmal segment of ALSA (Figure 5). Under the diagnosis of ALSA-esophageal fistula, the patient received thoracic endovascular aortic repair (TEVAR) and left subclavian artery coiling (Figure 6). The fistula was successfully treated, and the patient was hemodynamically stabilized. However, the patient died of pneumonia and sepsis two months after the procedure.

## 3. Discussion

With the advances in CT techniques, detailed anatomies of aortic arch anomalies and their spatial relationships to adjacent organs are well displayed. The right aortic arch with ALSA is the most common variation of a right aortic arch [4,6]. According to the hypothetical double aortic arch theory [4], the anomaly is due to a regression of an arterial segment between the left common carotid and left subclavian arteries, as shown in Figure 1. The ALSA is the last branch of the arch. It takes an oblique retroesophageal course. Its proximal portion usually forms a Kommerell’s diverticulum, which is a remnant of the left arch. With the presence of ligamentum arteriosum connecting the diverticular portion of ALSA and the left pulmonary artery, this anomaly forms a vascular ring encircling the esophagus and the trachea. In this circumstance, the esophagus is compressed posteriorly by the diverticular portion of ALSA [7]. Esophageal compression may be aggravated in older individuals with progressed ectasia, tortuosity, and atherosclerotic change of the artery. The vascular pressure on the esophagus can create a fistula [8,9].

Previously, aortic arch anomalies have been reported as a miscellaneous etiology of aortoesophageal fistulae [1]. A review of previously reported cases of aberrant subclavian artery-esophageal fistulae revealed the prolonged use of devices such as NG and endotracheal (ET) tubes as a major contributing factor [2,5,10,11,12,13,14,15]. NG and ET tubes are believed to create pressure necrosis and erosion on the area of the esophageal wall that is in contact with the tubes [2,11]. Thus, the prolonged use of NG and/or ET tubes in a patient with a retroesophageal aberrant artery or other aortic arch anomalies forming a vascular ring may make the patient susceptible to arterial-esophageal fistula formation. A literature search showed a total of 40 documented cases of aberrant subclavian artery and esophageal fistulae (Table 1) [14,15,16,17,18,19,20,21,22,23,24,25,26,27,28,29,30,31,32,33,34,35,36,37,38,39,40,41,42,43,44]. Among those cases, a long-term use of NG and/or ET tube was thought to be the main cause of fistula formation in at least 18 cases. Although some cases were associated with esophageal cancer or recent procedures such as surgery and stent insertion, fistulae could be attributable to a prolonged use of NG and/or ET tubes in more cases. Therefore, some authors recommend screening intensive care patients and avoiding the long-term use of NG or ET tubes in patients with an aberrant subclavian artery [12,13]. Since conventional surgical repair takes considerable time, endovascular treatment such as temporary balloon occlusion or stent insertion is commonly performed for rapid bleeding control [29,30,31,32].

In the present case, we overlooked and misjudged several points that physicians should keep in mind. First, the presence of an arch anomaly was not taken seriously. Although the right aortic arch was suggested on the chest radiograph at the first visit, we did not immediately evaluate the relationships between major branches of aortic arch and the esophagus. Second, we did not treat the patient when he presented with minor bleeding. We only performed TEVAR and coiling procedures when the patient had massive bleeding during a revisit. According to our review of documented cases of aberrant subclavian artery and esophageal fistulae so far, 24 of 40 patients (60%) eventually died of massive bleeding or associated complications. Since sentinel hemorrhage is present in majority cases of aortoesophageal fistula before life-threatening bleeding, a high index of suspicion and early intervention are important for favorable prognosis [1,5,10]. Lastly, not knowing the presence of adjacent ALSA, a biopsy of the esophageal ulcer was performed for the patient. It might have produced a devastating result during the procedure.

In conclusion, we report a patient with massive upper gastrointestinal bleeding secondary to ALSA-esophageal fistula after a prolonged use of an NG tube. Our report aims to raise awareness to the fact that the presence of an arch anomaly is potentially responsible for fatal hemorrhage associated with a simple medical device such as an NG tube in a chronically ill patient.

## Figures and Tables

**Figure 1 diagnostics-11-00195-f001:**
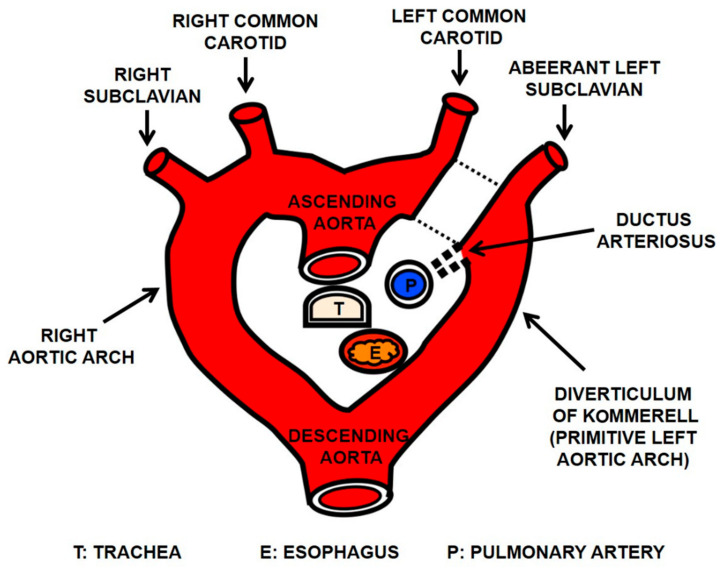
Diagram explaining the right aortic arch with aberrant left subclavian artery (ALSA) arising from a diverticulum of Kommerell based on the double aortic arch hypothesis. This anomaly results from a regression of the left arch between the left common carotid and left subclavian arteries (dotted lines). The left subclavian artery arises as the last branch of right aortic arch (ALSA) and takes a retroesophageal course. A remnant of the left arch becomes a diverticulum of Kommerell from which the normal-sized ALSA continues. With the presence of ligamentum arteriosum, a vascular ring is formed. The esophagus is compressed posteriorly by the diverticular portion of the ALSA.

**Figure 2 diagnostics-11-00195-f002:**
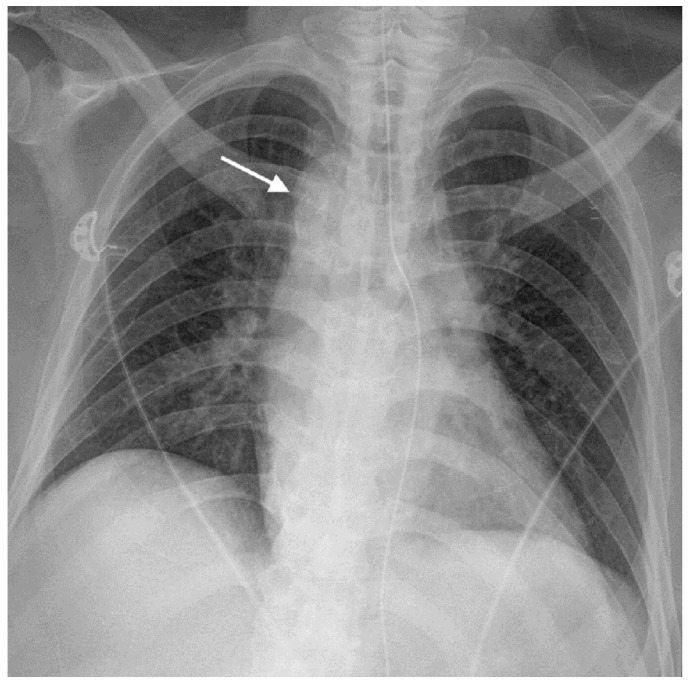
The radiograph with the anteroposterior view of the chest showed a right aortic arch (arrow).

**Figure 3 diagnostics-11-00195-f003:**
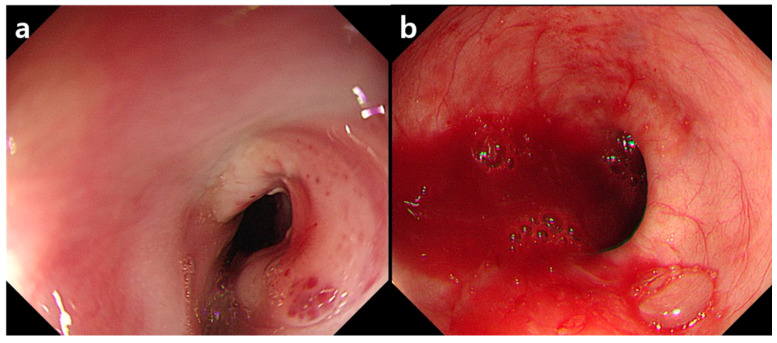
(**a**) Endoscopy on the first day showing a mass like lesion with an ulcer and red pigmentation in the upper esophagus. Active bleeding focus was not found. (**b**) After two days, massive blood pumping was seen in the ulcer base. A pulsating vessel was also noted (not shown here).

**Figure 4 diagnostics-11-00195-f004:**
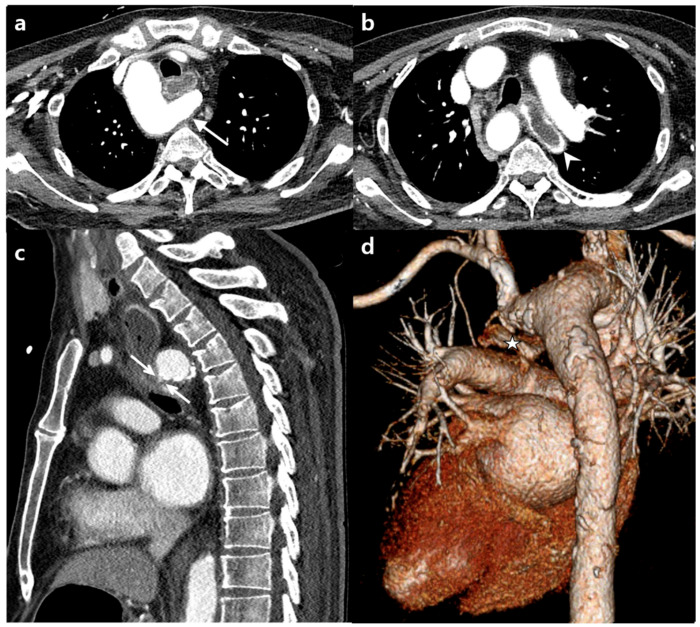
(**a**) Axial chest CT scans taken at the second visit showing the right aortic arch and retroesophageal aberrant left subclavian artery (ALSA, diverticular portion, arrow). (**b**) Note the contrast filling in the elongated esophagus (arrowhead). (**c**) Sagittal reformation image showing focal contrast extravasation from the diverticular portion of the ALSA into the esophagus (double arrows), possibly indicating an arterial-esophageal fistula. (**d**) Three-dimensional volume rendering image showing the relationship between the esophagus and the aberrant left subclavian artery. The esophagus (star) is compressed posteriorly and inferiorly by the diverticular portion of the ALSA.

**Figure 5 diagnostics-11-00195-f005:**
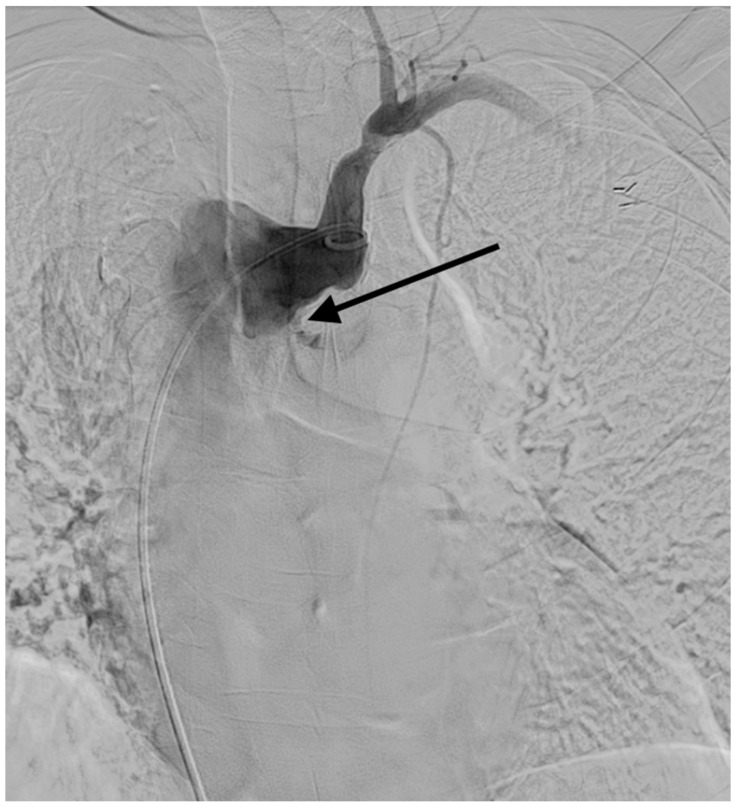
The arteriogram obtained after contrast injection at the ostium of the left subclavian artery confirmed the extravasation (arrow) from the diverticular segment of the aberrant left subclavian artery.

**Figure 6 diagnostics-11-00195-f006:**
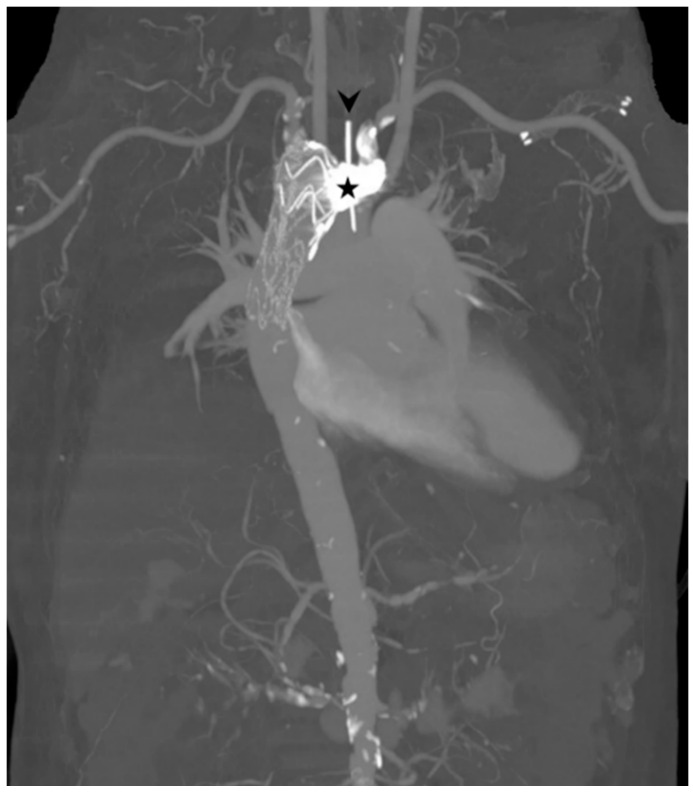
Maximum-intensity-projection image showing the result of the thoracic endovascular aortic repair and left subclavian artery coiling (star). An arrowhead indicates the nasogastric tube in the esophagus.

**Table 1 diagnostics-11-00195-t001:** Summary of reported cases of aberrant subclavian artery-esophageal fistulae.

References	Sex	Age (year)	Causes	ETT *	NGT ^†^	Endoscopic Findings	Treatment	Outcome	Sentinel Bleed
Lynn (1969) [44]	M	57	ARSA aneurysm	N	N	NIA	S	Died	NIA
Reynes et al. (1976) [43]	F	72	ARSA aneurysm	N	N	N	N	Died	Y
Merchant et al. (1977) [7]	F	17	NGT	N	Y	NIA	N	Died	NIA
Livesay et al. (1982) [11]	M	25	ETT/NGT	Y	Y	Mucosal defect in upper esophagus	B & S	Died	N
Jungck and Puschel (1983) [14]	M	6	ETT/NGT	Y	Y	NIA	B	Died	N
Belkin et al. (1984) [17]	M	27	NGT	N	Y	Fungating lesion on posterior wall of esophagus	B & S	Died	N
Edwards et al. (1984) [10]	M	79	ARSA aneurysm	N	N	Extrinsic compression at posterior wall of the esophagus	S	Died	Y
	F	36	NGT	Y	Y	Active hemorrhage in the esophagus	N	Died	Y
Gossot et al. (1985) [42]	F	72	ETT/NGT	Y	Y	NIA	NIA	Died	NIA
Guzzetta et al. (1989) [22]	F	4 mo	NGT	Y	Y	N	S	Survived	Y
Kullnig (1989) [40]	M	66	ARSA aneurysm	N	N	N	N	Died	Y
Stone et al. (1990) [31]	M	72	ETT	Y	NIA	N	S	Died	N
Ikeda et al. (1991) [15]	M	9	ETT/NGT	Y	Y	NIA	NIA	Died	NIA
Hirakata et al. (1991) [23]	M	55	NGT	NIA	Y	N	B	Survived	N
Warshauer et al. (1993) [38]	M	73	ALSA aneurysm	Y	NIA	Extrinsic compression of esophagus	S	Survived	Y
Miller et al. (1996) [33]	F	11	ETT/NGT	Y	Y	Focal bleeding in the esophagus	B & S	Survived	N
Singha et al. (1998) [35]	M	82	ARSA aneurysm	N	N	Bleeding from fungating mass in upper esophagus	N	Died	N
Minyard and Smith (2000) [41]	F	39	NGT	NIA	Y	Normal esophagus	N	Died	NIA
Feugier et al. (2002) [20]	M	24	ETT/NGT	Y	Y	NIA	S	Survived	N
Eynden (2006) [19]	F	9	Esophageal stent	NIA	NIA	Blood oozing around the prosthesis	S & V	Survived	Y
Lehmann et al. (2006) [27]	M	78	Rupture of an ARSA aneurysm	N	N	Necrotic polypoid lesion in upper esophagus	B	Died	N
Milar et al. (2007) [2]	M	57	Esophageal carcinoma with esophagectomy and grastric pull-up	N	Y	A small anastomotic ulcer	S & V	Died	Y
Inman et al. (2008) [25]	M	63	Salivary bypass tube	NIA	Y	Bleeding focus not identified	S, V, & B	Died	Y
Magagna et al. (2008) [29]	F	73	Laryngeal carcinoma with prolonged radiotherapy	Y	N	Esophagitis and blood clots	B &V	Survived	Y
Fuentes et al. (2010) [21]	F	3	Esophageal prosthesis insertion	NIA	NIA	N	B, V, & S	Survived	N
Chapman et al. (2010) [18]	F	34	NGT	Y	Y	Active arterial bleeding at 25 cm from the incisors	B & S	Died	N
Situma et al. (2011) [36]	F	5 mo	Colonic interposition for esophageal atresia with a distal fistula	Y	Y	Blood clots at the colon-gastric junction	S	Survived	N
Jain et al. (2012) [26]	F	57	ETT/NGT	Y	Y	Vigorous bleeding from ARSA–esophageal fistula	B & S	Survived	N
Pop et al. (2012) [34]	M	67	Operation for esophageal cancer	NIA	N	Arterioesophageal Fistula	B & V	Died	N
Takahashi et al. (2013) [37]	M	63	ARSA aneurysm	N	N	Esophageal ulcer covered with fibrinous tissue	V &S	Survived	Y
Lo et al. (2013) [28]	NIA	16 mo	Stent insertion for esophageal atresia	N	Y	NIA	B, V, &	Died	N
		15 mo		N	Y	NIA	S	Survived	Y
Morisaki et al. (2014) [32]	F	74	Endovascular repair for ARSA aneurysm	NIA	NIA	esophageal mucosal erosion and fistula	B, V, & S	Died	N
Hosn et al. (2014) [24]	F	29	Esophageal stent	N	Y	Teflon patch	S	Survived	N
Joynt and Grifka (2015) [9]	NIA	17 mo	Spontaneous fistula	N	N	Small mucosa abnormality with white plaque	V & S	Survived	Y
Oliveira et al. (2016) [5]	M	20	ETT/NGT	Y	Y	Esophageal ulcer with copious bleeding	S	Survived	N
Kudose et al. (2017) [16]	F	20	ETT/NGT	Y	Y	NIA	N	Died	N
Shires et al. (2018) [13]	F	41	Tracheostomy tube/NGT	Y	Y	N	V & S	Died	Y
Zheng et al. (2019) [39]	M	67	Esophageal stent	NIA	NIA	N	V	Died	N
Merlo et al. (2020) [30]	F	29	Esophageal stent	Y	Y	Esophageal perforation	V & S	Survived	N

ETT: endotracheal tube, NGT: nasogastric tube, * Numbers in parentheses indicate the duration of ETT placement. ^†^ Numbers in parentheses indicate the duration of NGT placement. ARSA: aberrant right subclavian artery, ALSA: aberrant left subclavian artery, S: surgical repair, B: balloon temponade, V: endovascular repair, NIA: no information available, N: No, Y: Yes.

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
