# Peer review of "Aberrant Left Subclavian Artery-Esophageal Fistula in a Patient with a Prolonged Use of Nasogastric Tube: A Case Report and Literature Review"

_diagnostics, 2021, doi:10.3390/diagnostics11020195_

Round 1

Reviewer 1 Report

This article reported a patient with massive gastrointestinal bleeding secondary to aberrant left subclavian artery-esophageal fistula after a prolonged use of nasogastric tube. The author reviewed associated literatures and addressed a few critical issues that physicians should be aware of. The article is well written and provide informative messages to physicians. I hope my suggestions can help improve the article:

Major comments:

  1. It would be better if the author can add the chest radiograph suggesting right aortic arch at the first visit in Fig. 2.
  2. Did the patient receive temporary balloon occlusion or stent insertion? It would be better if author can add images after thoracic endovascular aortic repair and left subclavian artery coiling in Fig. 4.
  3. It’s good that the author did a thorough review of literature reporting patients with aberrant subclavian artery-esophageal fistula. It would be better if the author can add a column in Table 1 to describe the treatments of this disease. In addition, the outcome seemed to be poor in this disease, can you roughly estimate the mortality rate in the context?
  4. Esophagogastroduodenoscopy is usually the first diagnostic equipment for patients with upper GI bleeding. Therefore, in addition to a history of abnormal aortic arch and NG/ETT insertion, I think the findings of esophagogastroduodenoscopy, such as “site of ulceration/mass” can also hint the presence of an arterio-esophageal fistula. Can you review “the findings of esophagogastroduodenoscopy” in the literatures and add the information to Table 1.

Minor comments:

  1. In Fig 4. Arteriogram obtained after contrast “infection” at ostium of left subclavian artery confirmed extravasation (arrow) from the diverticular segment of the aberrant left subclavian artery. Should the word “infection” be “injection”?

Author Response

Answers to Reviewers’ Comments and Suggestions

Reviewer 1

This article reported a patient with massive gastrointestinal bleeding secondary to aberrant left subclavian artery-esophageal fistula after a prolonged use of nasogastric tube. The author reviewed associated literatures and addressed a few critical issues that physicians should be aware of. The article is well written and provide informative messages to physicians. I hope my suggestions can help improve the article:  

Major comments:

  1. It would be better if the author can add the chest radiograph suggesting right aortic arch at the first visit in Fig. 2.

à Thank you for your suggestion. We have added the initial chest radiograph (Fig 2) in page 4.

  1. Did the patient receive temporary balloon occlusion or stent insertion? It would be better if author can add images after thoracic endovascular aortic repair and left subclavian artery coiling in Fig. 4.

à The patient received endovascular repair using stent graft and coil embolization of ALSA. We have added the post-treatment image (Fig 6 in page 6).

  1. It’s good that the author did a thorough review of literature reporting patients with aberrant subclavian artery-esophageal fistula. It would be better if the author can add a column in Table 1 to describe the treatments of this disease. In addition, the outcome seemed to be poor in this disease, can you roughly estimate the mortality rate in the context?

à We have added a column for treatment in Table 1. According to our review of documented cases of aberrant subclavian artery and esophageal fistulae so far, 24 of 40 patients (60%) eventually died of massive bleeding or associated complications (page 10, line164).

  1. Esophagogastroduodenoscopy is usually the first diagnostic equipment for patients with upper GI bleeding. Therefore, in addition to a history of abnormal aortic arch and NG/ETT insertion, I think the findings of esophagogastroduodenoscopy, such as “site of ulceration/mass” can also hint the presence of an arterio-esophageal fistula. Can you review “the findings of esophagogastroduodenoscopy” in the literatures and add the information to Table 1.

à Thank you for your suggestion. We have added endoscopic findings in Table 1.   

Minor comments:

  1. In Fig 4. Arteriogram obtained after contrast “infection” at ostium of left subclavian artery confirmed extravasation (arrow) from the diverticular segment of the aberrant left subclavian artery. Should the word “infection” be “injection”?

à Thank you for pointing this out. We have corrected the typo.

Thank you for your insightful comments and suggestions, which helped us improve the quality of our manuscript significantly.

Reviewer 2 Report

Nice report. Great image quality.

The following changes would make the report read better.

Line 61- He was under hemodialysis treatment due to chronic renal failure- Please change to- -He was on hemodialysis for chronic renal failure.

Line 68- He could not take hemodialysis treatment due to a low blood pressure- Please change to- He could not get hemodialysis due to hypotension.

Line 71-On esophagogastroduodenoscopy- Please change to- Upper GI endoscopy

Line 70-Laboratory data showed a dropped hemoglobin- Please change to- Laboratory data demonstrated a drop in hemoglobin values.

Line 124 A review of documented cases involving aberrant subclavian artery-esophageal fistula has revealed an interesting contributing factor: a prolonged use of devices such as NG and endotracheal (ET) tubes- Please change to- A review of previously reported cases of aberrant subclavian artery-esophageal fistula revealed prolonged use of devices such as NG and endotracheal (ET) tubes as a major contributing factor.

Can you elaborate how ET tube can cause esophageal fistula?

Can you add a picture of the fistula treated with the endograft?

Is the illustration Fig1 original or you may have to get permission for use.

Author Response

Answers to Reviewers’ Comments and Suggestions

Reviewer 2

Comments and Suggestions for Authors

Nice report. Great image quality.

The following changes would make the report read better.

Line 61- He was under hemodialysis treatment due to chronic renal failure- Please change to- -He was on hemodialysis for chronic renal failure.

Line 68- He could not take hemodialysis treatment due to a low blood pressure- Please change to- He could not get hemodialysis due to hypotension.

Line 71-On esophagogastroduodenoscopy- Please change to- Upper GI endoscopy

Line 70-Laboratory data showed a dropped hemoglobin- Please change to- Laboratory data demonstrated a drop in hemoglobin values.

Line 124 A review of documented cases involving aberrant subclavian artery-esophageal fistula has revealed an interesting contributing factor: a prolonged use of devices such as NG and endotracheal (ET) tubes- Please change to- A review of previously reported cases of aberrant subclavian artery-esophageal fistula revealed prolonged use of devices such as NG and endotracheal (ET) tubes as a major contributing factor.

à Thank you for your suggestion. We revised the sentences according to your suggestion (page 3, line 62, 63, 64, 70, 72, and 73; page 7, line 134).

Can you elaborate how ET tube can cause esophageal fistula?

à According to literatures, there can be an erosion of the tracheal and esophageal wall by the continuous pressure between the endotracheal tube and the esophageal wall, particularly in the presence of a nasogastric or feeding tube within the esophageal lumen. Please take into account the references below and references no. 2 and 11 in our manuscript.

  • Hazy RC. Complications of the endotracheal tube following initial placement: Prevention and management in adult intensive care unit patients. https://www.uptodate.com/contents/complications-of-the-endotracheal-tube-following-initial-placement-prevention-and-management-in-adult-intensive-care-unit-patients
  • Bibas BJ, et al. Ann Transl Med. 2018;6:210. Surgery for intrathoracic tracheoesophageal and bronchoesophageal fistula

Can you add a picture of the fistula treated with the endograft?

à Thank you for your suggestion. We have added the post-treatment image in Fig 6 (page 6).

Is the illustration Fig1 original or you may have to get permission for use.

à Figure 1 is our original work.

Thank you for your insightful comments and suggestions, which helped us improve the quality of our manuscript significantly.